# Chromoblastomycosis in French Guiana: Epidemiology and Practices, 1955–2023

**DOI:** 10.3390/jof10030168

**Published:** 2024-02-22

**Authors:** Julie Valentin, Geoffrey Grotta, Thibaut Muller, Pieter Bourgeois, Kinan Drak Alsibai, Magalie Demar, Pierre Couppie, Romain Blaizot

**Affiliations:** 1Department of Dermatology, Andrée Rosemon Hospital, 97306 Cayenne, French Guiana; 2Department of Pathology, Center of Biological Resources (CRB Amazonie), Andrée Rosemon Hospital, 97306 Cayenne, French Guiana; mohamed.drakalsibai@ch-cayenne.fr; 3Laboratory of Parasitology, Andrée Rosemon Hospital, 97306 Cayenne, French Guiana; 4Tropical Biome and Immunophysiopathology (TBIP), Université de Lille, CNRS, Inserm, Institut Pasteur de Lille, U1019-UMR9017-CIIL-Centre d’Infection et d’Immunité de Lille, Centre Hospitalier de Cayenne, Université de Guyane, 97306 Cayenne, French Guiana

**Keywords:** chromoblastomycosis, endemic mycoses, neglected tropical diseases, French Guiana

## Abstract

Chromoblastomycosis (CBM) is a chronic neglected fungal disease, usually met in tropical areas. French Guiana is a South American territory with limited epidemiological data. This retrospective study concerned all patients with CBM proven by at least one paraclinical examination and diagnosed in French Guiana between 1950 and 2023. In total, 23 patients were included, mostly males (87%) of Creole origin, living in the coastal region (87%) and involved in outdoor occupations (74%). Lesions were mostly observed on the lower limbs (78.3%), with a median time to diagnosis of four years. Laboratory tests included positive direct microscopic examinations (78.3%) and mycological cultures (69.6%), identifying 14 cases of *Fonsecaea pedrosoi* and one case of *Exophiala janselmei.* Various treatments were employed, including antifungals, surgery and combinations of both. In conclusion, CBM in French Guiana involves a different population than other subcutaneous mycoses such as Lobomycosis or Paracoccidioidomycosis, mostly found in the forest hinterland. Surgery should be recommended for recent and limited lesions. Itraconazole and terbinafine should systematically be proposed, either in monotherapy or in combination with surgery or cryotherapy.

## 1. Introduction

Chromoblastomycosis (CBM) is a chronic deep mycosis, categorized as a Neglected Tropical Disease (NTD) by the World Health Organization [1]. This cutaneous and subcutaneous fungal infection is the second most frequent implantation mycosis in the world [1], caused by the traumatic inoculation of melanised (Dematiaceous) dimorphic fungi ubiquitous in the soil or plants. Several species may be involved, *Fonsecaea pedrosoi* and *Cladophialophora carrionii* being the most prevalent [2]. CBM can be found worldwide, but is usually met in tropical areas [3]. A recent study estimating the global burden of CBM showed that South America was the region with the highest worldwide prevalence [4]. The prevalence is higher among males, and it is considered an occupational disease as it often occurs among farmworkers, hunters or miners [1,5]. There may be a hormonal or a genetic susceptibility, especially in patients with the HLA-A29 allele [1,6,7]. Numerous studies have been published, notably in Asia (Japan and China [8]), Madagascar [9] and Latin America (mostly Brazil and Mexico) [10]. 

Once suspected, the diagnosis is relatively easy, based on a compatible clinical presentation, the occurrence in an endemic area and simple paraclinical examinations. Superficial sampling of the scales and microscopic examination after exposition to potassium hydroxide (KOH) can reveal pathognomonic fumagoid cells, also called muriform cells, sclerotic bodies or medlar bodies [11]. However, CBM is difficult to treat. The duration of the disease as well as the severity of the lesions are correlated with a greater resistance to treatment [12]. These treatments must therefore be prolonged, with a high risk of relapse when stopped. 

French Guiana is the only French overseas territory in South America, located between Surinam and Brazil. The first cases of CBM were reported in Brazil [13], a country which shares a 730 km border with French Guiana. This French territory is mostly covered by a sparsely populated rainforest. Most of the population dwells on the small coastal strip. The Andrée Rosemon Hospital in Cayenne (capital) is the only referral center for dermatology. Primary care in the forest hinterland is provided by remote health centers. French Guiana benefits from a French universal health care system. The last published work on CBM in French Guiana dates back to 2001 [14], and there is a deep lack of epidemiological data on this neglected disease in this unique setting of a French territory in the Amazon. 

Our aim was to describe the epidemiology of CBM in French Guiana as well as the evolution of clinical, diagnostic and therapeutic practices from 1955 to 2023.

## 2. Material and Methods

We conducted an observational, retrospective study at the Cayenne Hospital. We selected all patients with CBM proven by at least one paraclinical test (direct microscopic examination of skin scraping/mycological culture on skin biopsy/histopathology on skin biopsy) who consulted the Dermatology Department of Cayenne or the remote health centers of French Guiana between 1955 and 2023. Data were collected from medical records and databases of the mycology and histopathology laboratories. We analyzed demographic data, clinical characteristics, histological and microbiological findings, treatments used and clinical evolution. We used the clinical severity grades proposed by Queiroz-Telles [10] to establish the clinical severity of the disease, as follows: Mild (single plaque or nodule less than 5 cm);Moderate (one or more nodular lesions, plaque or verrucous less than 15 cm in one or two adjacent areas);Severe (lesion larger than 15 cm or several non-adjacent areas). Species identification was based on their morphological characters on positive fungal culture.

This study was conducted according to the principles of the Helsinki declaration. All data were analyzed anonymously. Collective information on this study was provided to all patients in the Dermatology Department. This study was authorized by the Direction for Research, Innovation and Public Health (DRISP) of the Cayenne Hospital Center. For this internal retrospective study and according to French law, no further legal clearance was necessary.

## 3. Results

### 3.1. Demography

During the study period, 23 patients met the inclusion criteria. Among them, 20 (87%) were men. The mean age at diagnosis was 60 years old [range 43–74]. The patients’ general characteristics are shown in Table 1. Most of them (20/23, 87%) lived in the coastal area of French Guiana, and only three came from the hinterland (Figure 1). Seventeen patients (74%) had an outdoor occupation. An initial trauma was found in nine patients (39.1%). 

### 3.2. Clinical Presentation

The median duration of CBM at diagnosis was 4 years [2 months–20 years], and 12 patients (52.2%) had lesions evolving for at least 3 years. Twelve patients (52.2%) had a single lesion. The lower limbs were mainly involved (78.3%), followed by the upper limbs (21.7%). One patient had both a leg and a scrotal lesion. Various clinical manifestations and associated symptoms were described (Figure 2, Table 2). Three patients (13%) experienced long-term complications. One presented functional limitation because of a lesion on his hand, and one experienced an elephantiasis on his affected leg. The last patient developed a squamous cell carcinoma with involvement of the underlying bone, requiring amputation. 

### 3.3. Laboratory Tests

Direct microscopic examination was positive in 12/23 patients (52.2%), showing fumagoid cells after KOH treatment. In addition, 16/23 patients (69.6%) had a positive mycological culture on skin biopsy (Figure 3). Among these 16 patients, *F. pedrosoi* was identified in 14 patients. In the first patient included, whose diagnosis was made in 1955, *Exophiala janselmei* was isolated in fungal culture. In one case, the species could not be determined. Of note, all identifications were done morphologically.

Histopathology was positive in 22/23 cases (95.7%), showing fumagoid cells either in micro-abcsesses, in giant cells or isolated in skin tissue. The other patterns were dermal granuloma (16 cases, 72.7%), micro-abcesses (nine cases, 40.9%), epidermal hyperplasia (eight cases, 36.3%) and dermal sclerosis (four cases, 18.1%) (Figure 4 and Figure 5). 

### 3.4. Therapeutic Regimen

Various treatments were used, depending on the year of diagnosis. Patients received an average of 2.3 lines of treatment, and no less than 52 treatment regimens were proposed. 

Different antifungal treatments have been used over time, including oral 5-fluorocytosine (5-FC) alone or with topical 5-FC or itraconazole from 1972 to 1990. Dual therapy with itraconazole (200 to 400 mg daily) and 5-FC (12 to 24 tablets daily) was the most widely prescribed combination, administered to five patients between 1982 and 1990. One patient presented a relapse after three months, while two were switched to another treatment and the remaining two were lost to follow-up. Furthermore, between 1987 and 1988, itraconazole was prescribed as the only treatment for two patients. One was cured but relapsed after 18 months of treatment, and one had a partial remission. Itraconazole has also been proposed as a maintenance treatment following a combination of 5-FC and itraconazole in one patient till complete response. Additionally, oral terbinafine (250 to 1000 mg/d) was administered alone or with cryotherapy in eight patients between 2001 and 2023. Two patients experienced clinical improvement, one is cured and off treatment, two are still undergoing treatment, and three patients were lost to follow-up.

Some older antifungal agents, like potassium iodide and miconazole infusion, or combination therapy associated with isoniazid, sulfamethoxypyridazine and weekly fungizone infiltrations or thiabendazole, were used between 1969 and 1978 but were found to be poorly tolerated and ineffective. 

A monotherapy with surgery was proposed to four patients. For three patients, lesions were recent (evolving for 2–3 months) and surgery was the first-line treatment. One of them was cured, while the two others were lost to follow-up. The last one had a 5-year evolving CBM over the whole lower limb. She experienced a relapse after surgery, probably because no adjuvant antifungals were prescribed. Three other patients were proposed surgery in association with a systemic antifungal agent. We recorded one complete cure, one partial response and one lost to follow-up. 

### 3.5. Outcomes

Data concerning the last treatment used for each patient and the outcome at the last visit are presented in Table 3. However, these data are difficult to analyze, as several patients were lost to follow-up after a visit showing a partial cure. Patients who were immediately lost to follow-up without any visit are presented in the “Lost to follow-up” row. Furthermore, 21/23 patients received more than one line of treatment, and the final cure was reached after a succession of different therapeutic options. Though 13 patients received antifungals without surgery or cryotherapy as a final treatment, several lines of different antifungals were sometimes used successively. Itraconazole (200 mg to 400 mg daily), 5-FC (12 to 24 tablets daily) and terbinafine (1 g daily) were the most frequent antifungals. At the cut-off date in 2023 (which was sometimes very distant from the last visit), 11 patients were lost to follow-up (47.8%), nine patients were clinically cured at the last follow-up visit (39.1%), and three patients were still undergoing treatment (13.1%).

## 4. Discussion

We propose here a global view of the epidemiology of CBM in French Guiana between 1950 and 2023. The Cayenne Hospital Center is the only referral center for dermatology in French Guiana, and all cases of chromoblastomycosis are referred to its dermatological team. Therefore, we can assume that all cases of chromoblastomycosis diagnosed in French Guiana during the study period were included in our databases. However, it is still possible that patients remained undiagnosed due to barriers to health care access. Some of these cases have already been published [15,16,17], while most remained unreported until now. 

Our study population was consistent with the literature: all patients were adults, with a male predominance (ratio 7:1), occupations mainly focused on agriculture and outdoor work [5,18] and lesions mainly involving the lower limbs [4]. There was a significant delay in diagnosis, with a median of 4 years, which can be explained by several factors. Isolation and socio-economic hardships are major barriers to health care access in French Guiana. Indirect costs such as transportation in tropical remote areas can limit the patient’s care-seeking behavior. Moreover, the slow, insidious and initially asymptomatic evolution of lesions can delay patients whose professions and outdoor activities result in frequent benign limb injuries [19]. The multiplicity of differential diagnoses [20] and the rarity of the disease can make clinical recognition of the disease difficult for physicians unfamiliar with it [4]. Dermoscopy can be a useful diagnostic tool [21,22]. 

Apart from our first patient in whom *E. janselmei* was identified, *F. pedrosoi* was the only species found, already known as the most prevalent in South America and the Amazon [3,5,23]. Indeed, it seems to be associated with humid areas [2]. Scraping the lesions for direct examination after KOH treatment is easy and cost-effective when health professionals are trained [4]. Histological analysis of skin biopsies remained the most important diagnostic test. Diagnostic confirmation is easy in histology, with several suggestive signs and one pathognomonic sign (observation of fumagoid bodies) [11]. However, the biopsy can be negative despite a positive direct examination, as the fungus is eliminated from the dermis through the stratum corneum [24]. In our study, pathological analysis was 91.3% effective for initial diagnosis. However, the absence of histopathology laboratories in remote areas probably limits the availability of early diagnosis. New diagnostic tests are based on the detection of specific biomarkers or DNA sequencing, but this raises the question of their accessibility in endemic countries with limited resources [25,26,27]. The proportion of diagnoses by direct examination was low, given the easiness of this procedure. This might be explained by the fear of missing a differential diagnosis leading to a systematic skin biopsy, or a lack of training in some dermatologists not aware of the clinical aspect of CBM or the most suitable mycological test.

The epidemiology of CBM in French Guiana differs from observations in other endemic deep mycoses such as lobomycosis or paracoccidioidomycosis. Indeed, the latter two are mostly observed among gold miners working in the rain forest hinterland [28] Most of these patients are Brazilian migrants working illegally in French Guiana. Conversely, our study shows that CBM in French Guiana occurs mainly in farmers and patients of Creole origin living on the coastal strip. Only one of our patients lived in the hinterland. It probably reflects the importance of agriculture in the coastal region, though we cannot explain why traditional agriculture in Amerindians and Maroons in the hinterland does not lead to CBM cases. The deep Amazon rainforest in the French Guiana hinterland might be less suited for CBM, though Lobomycosis and Paracoccidiodiomycosis are known to thrive in this environment.

As in lobomycosis, the treatment of CBM remains difficult and controversial. There is no consensus about management of CBM. Surgical management by excision into healthy margins should always be proposed as first-line treatment whenever possible, as it usually leads to healing, particularly for mild forms [10]. Mohs micrographic surgery can also be proposed, with excellent efficacy and no distant recurrence [29], though access to this technique is limited due to its low availability, high cost, and the requirement for trained and available teams. Surgery in association with systemic antifungal agents can also be proposed [30], though deep lesions or difficult locations could leave bone or tendon structures exposed and would be a contra-indication. However, delayed diagnosis often makes such management impossible. In our study, the only patient who experienced a relapse after surgery had a CBM evolving for 5 years, and we can suppose that margins were not clear. Surgery was performed in three other patients who consulted early and presented a mild form, enabling them to be cured without recurrence after the procedure. 

In moderate to severe forms, general antifungal agents seem to be the first choice, although their results vary. In vitro tests showed susceptibility of *Fonsecaea* species to terbinafine, voriconazole [23,31], itraconazole [23] or ketoconazole [32], though the difficulty of in vitro cultures makes it impossible to standardize sensitivity tests [12]. In some studies, CBM caused by *Fonsecaea* showed a greater response to itraconazole than *Cladophialaphora* strains, but data are contradictory [8,33]. Dual therapy with itraconazole and 5-FC has shown good results, although the toxicity of 5-FC limits its use today [34,35]. Dual therapy with itraconazole and terbinafine may also be interesting in refractory CBM due to their synergistic effect [33], as well as posaconazole [34]. Cryotherapy, local heat therapy or topical imiquimod are interesting adjuvant therapies [10,35]. As the literature becomes more extensive, new alternatives are emerging. HIV peptidase inhibitors are reported to disrupt *F. pedrosoi* peptidase secretion and development, modify its structure and virulence and increase the capacity of macrophages to eliminate it [36,37]. Inhibitors of protein kinases such as genistein and stauporine may prevent fungus invasion [38]. Interest in photodynamic therapy (PDT) has been growing for several years as an alternative to conventional treatments for CMB or as an adjuvant therapy. One study showed its in vitro efficacy on *F. pedrosoi* and *C. carrionii*. 5-aminolevulinic acid PDT (ALA-PDT) has been used in combination with systemic antifungal agents in refractory CMB with clinical improvement after four to nine sessions [39], or with isotretinoin and CO2 laser in a patient with hyperkeratotic lesions with clinical success [40]. These new therapies appear to be of interest, being less toxic than systemic antifungal agents, less invasive than surgery and offering rapid clinical improvement. 

In light of these data, we propose a therapeutic algorithm (Figure 6). Today, terbinafine and itraconazole, alone or in combination, appear to be the systemic antifungal agents of first choice.

Though several therapeutic options are available, CBM is susceptible to relapse and a long and cautious follow-up is necessary before final cure can be established before asserting a full cure. Our practices have not been consistent over time. In the 1980s, a patient was considered cured when the direct examination of skin scales was negative, and there were no recurrences within 5 years. Eventually, patients were often lost to follow-up before the end of the monitoring period. Today, our patients are declared cured solely based on clinical criteria. However, Queiroz-Telles et al. suggested that the decision to discontinue treatment required a 2-year recurrence-free follow-up period and should be based on clinical (resolution of lesions and symptoms), mycological (negative direct microscopic examination and culture) and histological criteria (absence of fumagoid cells) [10].

## 5. Conclusions

CBM is an NTD whose global burden is unclear due to a lack of solid epidemiological and therapeutic studies. This study highlights the epidemiological characteristics of CBM in French Guiana, which primarily affects coastal-dwelling individuals of Creole descent engaged in agriculture, in contrast to other endemic deep mycoses. The predominant pathogen was *F. pedrosoi*. Treatment approaches varied over the study period. Surgery should be recommended as the initial treatment for recent and localized lesions, while first-line antifungal therapies, such as itraconazole and terbinafine, can be combined for severe or resistant cases. Given the potential for relapse, continued monitoring post-treatment is essential. 

## Figures and Tables

**Figure 1 jof-10-00168-f001:**
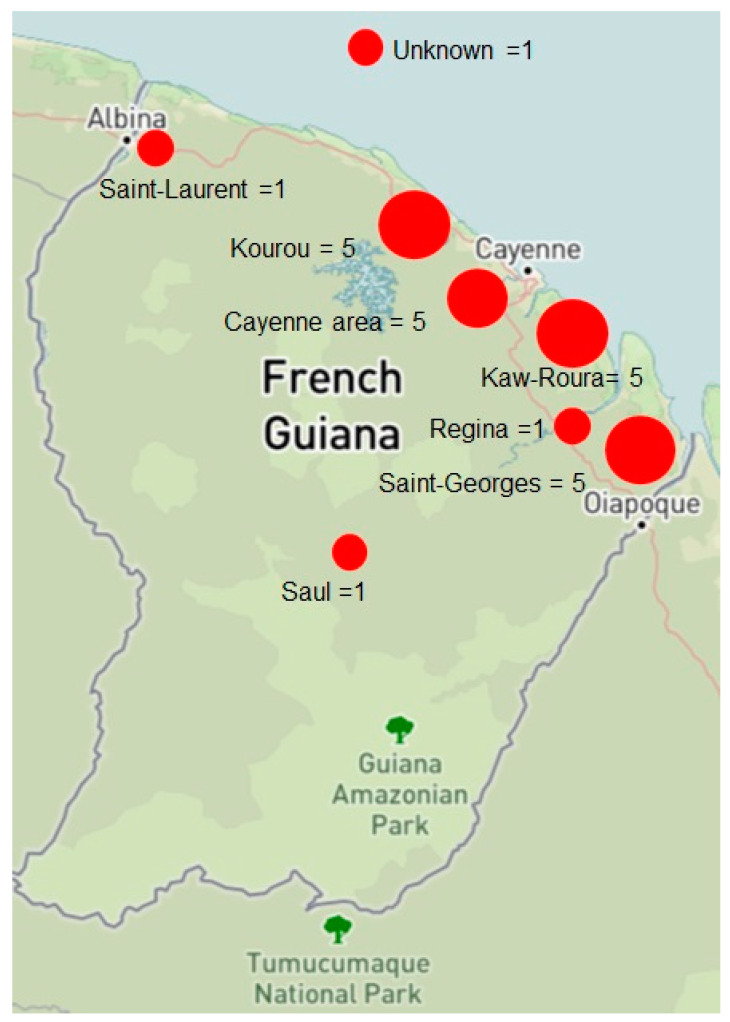
Likely place of contamination of CBM cases, 1955–2023, French Guiana (adapted from https://ngmdb.usgs.gov/topoview accessed on 12 February 2024).

**Figure 2 jof-10-00168-f002:**
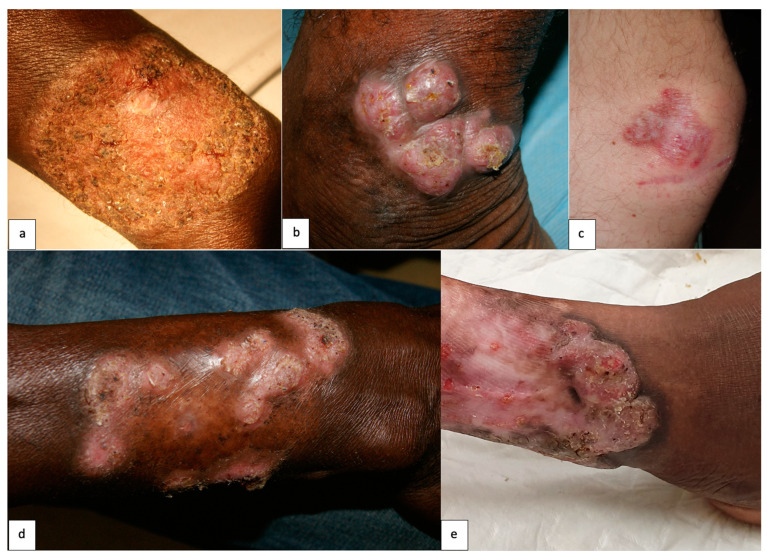
Clinical manifestations of chromoblastomycosis, French Guiana, 1955–2023. (**a**) Verrucous lesion on the wrist. (**b**) Nodular lesions on the ankle. (**c**) Soft erythematous plaque lesion on the knee. (**d**) Cicatricial lesion with active verrucous peripheral border and central scarring. (**e**) Mixed lesion of the leg with a proximal cicatricial part and a distal tumoral part.

**Figure 3 jof-10-00168-f003:**
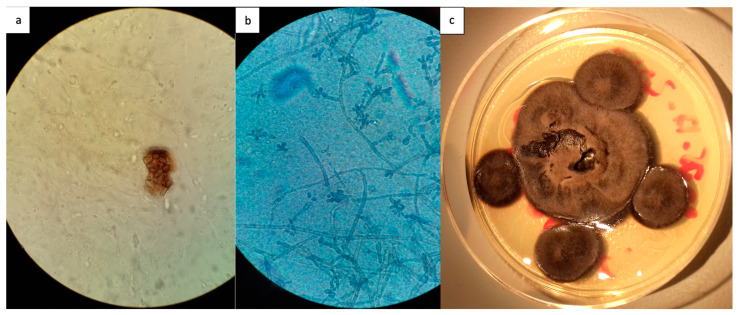
Diagnostic laboratory tests of chromoblastomycosis, French Guiana, 1955–2023. (**a**) Direct examination showing fumagoid cells after potassium hydroxide treatment, ×10 (**b**) Direct examination after digestion of a fresh skin biopsy fragment. Presence of mycelial filaments and fumagoid cells, ×40. (**c**) Mycological culture of a pigmented black fungus corresponding to *Fonsecaea pedrosoi*.

**Figure 4 jof-10-00168-f004:**
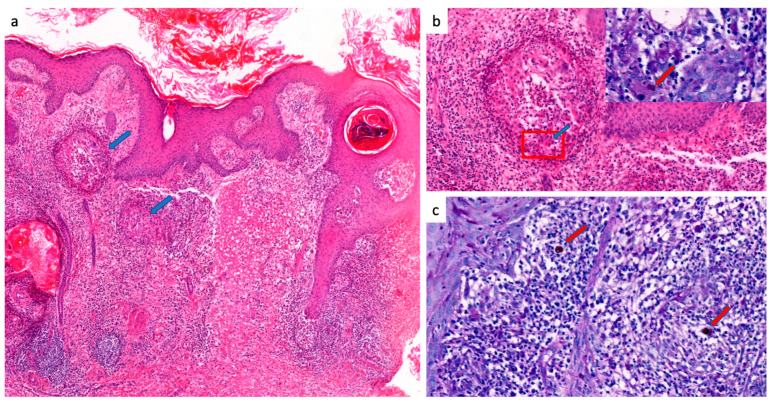
Histological patterns of chromoblastomycosis, French Guiana, 1955–2023 (part 1) (**a**) Reactive pseudoepithelimatous epidermal hyperplasia with hyperkeratosis. The entire dermis is covered by a dense polymorphic inflammatory infiltrate including some granulomas (blue arrows) (Hematoxylin and Eosin “H&E” stain, 200× magnification). (**b**) The inflammatory infiltrate consists of numerous neutrophils and eosinophils in the center, associated with sheets of histiocytes, epithelioid cells, lymphocytes and multinucleated giant cells at the periphery (blue arrow and red box) (H&E stain, 400× magnification). Presence of intracellular dark brown bodies within the multinucleated giant cell, corresponding to chromoblastomycosis bodies (red arrow) (Periodic Acid Schiff “PAS” stain, 100× magnification). (**c**) This image shows numerous chromoblastomycosis bodies in the form of rounded parasitic phase cells measuring between 5 and 12 µm arranged in muriform structures also called “copper pennies” (red arrows) (PAS stain, 600× magnification).

**Figure 5 jof-10-00168-f005:**
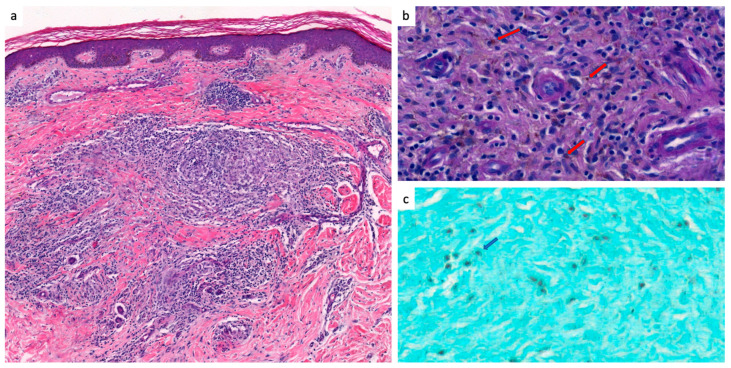
Histological patterns of chromoblastomycosis, French Guiana, 1955–2023 (part 2) (**a**) The epidermis is discreetly acanthosic and slightly hyperkeratotic. The dermis shows an inflammatory infiltrate consisting of epithelioid and histiocytic granulomas associated with some multinucleated giant cells (HE stain, 100× magnification). (**b**) Within the granulomas, presence of small isolated round chromoblastomycosis parasitic phase cells (red arrows) (PAS stain, 1000× magnification). (**c**) Presence of some dark-stained parasitic phase cells of chromoblastomycosis highlighted by Gomori-Grocott stain (blue arrow) (Gomori-Grocott stain, 1000× magnification).

**Figure 6 jof-10-00168-f006:**
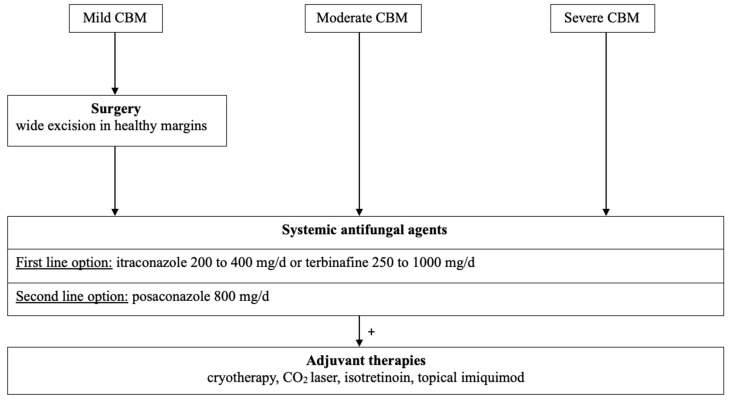
Therapeutic algorithm for chromoblastomycosis, French Guiana, 2023.

**Table 1 jof-10-00168-t001:** General characteristics of the study population, French Guiana, 1955–2023.

Characteristics	Number of Patients *n* (%)
Gender ● Female ● Male	● 3 (13) ● 20 (87)
Median age at diagnosis	60 (43–74)
Ethnicity ● Creole ● Haitian ● Surinamese ● Maroon ● Brazilian ● French (mainland) ● Saint-Lucian ● Unknown	● 11 (47.8) ● 3 (13) ● 2 (8.7) ● 1 (4.3) ● 1 (4.3) ● 1 (4.3) ● 1 (4.3) ● 3 (13)
Occupation ● Farmer ● Gold digger ● Warehouseman ^a^ ● Forestry worker ● Gardener ● Hunter ● Ornithologist ● Mason ● Unknown	● 10 (43.3) ● 2 (8.7) ● 2 (8.7) ● 1 (4.3) ● 1 (4.3) ● 1 (4.3) ● 1 (4.3) ● 1 (4.3) ● 4 (17.4)
Likely place of contamination ● Kaw and Roura ● Kourou area (Montsinéry) ● Saint-Georges-de-l’Oyapock ● Cayenne urban area (Matoury, Macouria) ● Saint-Laurent-du-Maroni ● Saül ● Regina ● Unknown	● 5 (21.7) ● 5 (21.7) ● 5 (21.7) ● 4 (17.4) ● 1 (4.3) ● 1 (4.3) ● 1 (4.3) ● 1 (4.3)
Underlying condition ● Digestive parasitosis ^b^ ● Cutaneous leishmaniasis ● Vascular (high blood pressure, stroke) ● Prostate carcinoma ● None ● Unknown	● 6 (26.1) ● 2 (8.7) ● 2 (8.7) ● 1 (4.3) ● 3 (13) ● 11 (47.8)

^a^ One of the warehousemen hunted and cultivated a vegetable garden as a hobby; ^b^ a digestive parasitosis was also found in one of the patients with a cutaneous leishmaniasis and in the patient with the prostate carcinoma.

**Table 2 jof-10-00168-t002:** Clinical presentation of chromoblastomycosis, French Guiana, 1955–2023.

Clinical Presentation	Number of Patients (%)
Lesion type ● Cicatricial ● Verrucous ● Nodular ● Tumoral ● Plaque ● Mixed	● 5 (21.7) ● 5 (21.7) ● 3 (13) ● 1 (4.3) ● 1 (4.3) ● 8 (34.8)
Clinical severity ● Mild ● Moderate ● Severe	● 3 (13) ● 12 (52.2) ● 7 (30.4)
Associated symptoms ● Pruritus ● Pain ● Edema ● Lymphadenitis ● Functional limitation	● 5 (26.1) ● 4 (17.4) ● 2 (8.7) ● 1 (4.3) ● 1 (4.3)

**Table 3 jof-10-00168-t003:** Therapeutic outcomes according to the last treatment used for each patient with CBM, French Guiana, 1955–2023.

Outcome	Antifungals Alone *n* = 13	Antifungals + Surgical Excision (*n* = 3)	Antifungals + Cryotherapy (*n* = 3)	Unknown (*n* = 4)
Cured	5 (38.5%)	2 (66.7%)	2 (66.7%)	-
Partially cured	5 (38.5%)	-	1 (33.3%)	-
Not cured	1 (7.7%)	-	-	-
Lost to follow-up	2 (15.4%)	1 (33.3%)	-	4

## Data Availability

Data are contained within the article.

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
