# Peer review of "Chromoblastomycosis in French Guiana: Epidemiology and Practices, 1955–2023"

_jof, 2024, doi:10.3390/jof10030168_

Round 1
Reviewer 1 Report
Chromoblastomycosis is a relevant endemic mycosis, It is important in South America, and it is necessary to know its real prevalence on this continent. Although the topic addressed in this way is not new, there is a need for it to be published.
However, I have some considerations that should be clarified
1.Table 3:
I was a little confused by the description of the treatments. I understand that patients had several patients and that, therefore, in the end, the number of patients treated is much greater than the number of patients in the study, but I believe that the authors can improve this presentation, making the reader understand when reading the table, that this is happening. Furthermore, in the surgical excision item there are 4 procedures, but in the results, there are 4 cures and 1 recurrence, resulting in 5 procedures.
2. The authors state that the cases of chromoblastomycosis diagnosed in French Guiana would all have passed through that dermatological service, however, we have published articles on cases diagnosed in 1969 by Pradinaud. Since the authors are proposing a global study, wouldn’t it be important to review published cases from French Guiana?
3. The therapeutic algorithm for chromoblastomycosis can’t be extended globally. Improvement, for example, with terbinafine, is not a rule, despite showing sensitivity (?) in vitro. Surgical excision of mild lesions is also not recommended for all lesions. There are injuries on the extremities (hands and feet), which would leave bone or tendon structures exposed after surgery with expansion of the limits. I think this should be exposed in this proposal.
In the description of treatment, due to the existence of many treatments over the years, I realized that it needs better clarity. Also clarify whether all diagnosed cases are actually in this review, as there are reports in the literature that are not mentioned.
Paper published about Chromoblastomycosis in French Guiana
84.Silverie R, Ravisse P. On 2 cases of chromomycosis observed in French Guiana. Bull Soc Pathol Exot Filiales. 1962 Sep-Oct; 55:751–2. PMID: 13992936
85. Pradinaud R, Joly F, Basset M, Basset A, Grosshans E. Chromomycosis and Jorge Lobo disease in French Guiana. Bull Soc Pathol Exot Filiales. 1969 Nov-Dec; 62(6):1054–63. PMID: 5409181
86. Pradinaud R. Traitement de 6 chromomycoses par la 5 fluorocytosine em Guyane franc¸aise. Nouv Presse Med. 1974; 3(31):1955. PMID: 4444938
Author Response
Reviewer 1
Chromoblastomycosis is a relevant endemic mycosis, It is important in South America, and it is necessary to know its real prevalence on this continent. Although the topic addressed in this way is not new, there is a need for it to be published.
However, I have some considerations that should be clarified
1.Table 3:
I was a little confused by the description of the treatments. I understand that patients had several patients and that, therefore, in the end, the number of patients treated is much greater than the number of patients in the study, but I believe that the authors can improve this presentation, making the reader understand when reading the table, that this is happening. Furthermore, in the surgical excision item there are 4 procedures, but in the results, there are 4 cures and 1 recurrence, resulting in 5 procedures.
- We agree with the reviewer. This analysis is very complex, due to the large number of different therapeutic options used, and the succession of different antifungals before reaching a final cure in some patients. We entirely rewrote the Table 3 by providing the data at the last visit and summarizing the courses of antifungals. See Results, page 15, lines 220-228: “Data concerning the last treatment used for each patient and the outcome at the last visit are presented in Table 3. However, these data are difficult to analyze, as several patients were lost to follow-up after a visit showing a partial cure. Patients who were immediately lost to follow-up without any visit are presented in the “Lost to follow-up row”. Besides, 21/23 patients received more than one line of treatment, and the final cure was reached after a succession of different therapeutic options. Though 13 patients were treated with antifungals without surgery or cryotherapy, several lines of different antifungals were sometimes used successively.”
- The authors state that the cases of chromoblastomycosis diagnosed in French Guiana would all have passed through that dermatological service, however, we have published articles on cases diagnosed in 1969 by Pradinaud. Since the authors are proposing a global study, wouldn’t it be important to review published cases from French Guiana?
- We thank the reviewer for this comment. Indeed, our study includes cases previously published by Pradinaud et al. These articles are now mentioned in the Discussion, page 17, lines 243-244: “Some of these cases have already been published (1–3) while most remained unreported until now” and the references have been added.
- The therapeutic algorithm for chromoblastomycosis can’t be extended globally. Improvement, for example, with terbinafine, is not a rule, despite showing sensitivity (?) in vitro. Surgical excision of mild lesions is also not recommended for all lesions. There are injuries on the extremities (hands and feet), which would leave bone or tendon structures exposed after surgery with expansion of the limits. I think this should be exposed in this proposal.
- We agree with the reviewer and added the following sentences in the Discussion, page 19, lines 291 -292 “though deep lesions or difficult locations could leave bone or tendon structures exposed and would be a contra-indication”
In the description of treatment, due to the existence of many treatments over the years, I realized that it needs better clarity. Also clarify whether all diagnosed cases are actually in this review, as there are reports in the literature that are not mentioned.
Paper published about Chromoblastomycosis in French Guiana
- We thank the reviewer for these comments. These patients are included among the 23 cases of our study. These papers have been added to the references.
- We also rewrote the “treatment” paragraph of the Results section.
Reviewer 2 Report
Comments and Suggestions for Authors
It is an interesting work on chromoblastomycosis, in French Guiana, where there are few reports, which makes it important to complete the epidemiology of the disease.
It is understood that it is a report from many years ago (equally valuable), where the species could not be precisely identified; Therefore, it is important that the authors emphasize that the identification of F. pedrosoi was done morphologically because there are other species. How was Exophiala jeanselmei identified (the name of the fungus must be corrected throughout the text), because there are also other species of Exophiala that cause the disease.
I propose that you put a map to know the origin of the cases, that makes epidemiology easier.
Give an explanation (in the discussion) about the low percentage of diagnoses by direct examination, since this test is quick and effective, compared to histopathology.
Author Response
Reviewer 2
It is an interesting work on chromoblastomycosis, in French Guiana, where there are few reports, which makes it important to complete the epidemiology of the disease.
It is understood that it is a report from many years ago (equally valuable), where the species could not be precisely identified; Therefore, it is important that the authors emphasize that the identification of F. pedrosoi was done morphologically because there are other species. How was Exophiala jeanselmei identified (the name of the fungus must be corrected throughout the text), because there are also other species of Exophiala that cause the disease.
- We thank the reviewer for this interesting comment. Indeed, the identification of F. pedrosoi was made morphologically. Though mass spectrometry is now available at the Cayenne hospital, all cases were determined morphologically, including the only case of Exophiala janselmei, in 1955. Please see results, page 11, lines 160-161: “Of note, all identifications were done morphologically.
- Besides, the taxonomy of Exophiala janselmei has been corrected throughout the manuscript.
I propose that you put a map to know the origin of the cases, that makes epidemiology easier.
- We thank the reviewer for this suggestion. A map has been added as Figure 1.
Give an explanation (in the discussion) about the low percentage of diagnoses by direct examination, since this test is quick and effective, compared to histopathology.
- We completely agree with the reviewer, and added the following sentences in the Discussion, page 18, lines 268-272: “The proportion of diagnoses by direct examination was low, given the easiness of this procedure. This might be explained by the fear of missing a differential diagnosis leading to a systematic skin biopsy, or a lack of training in some dermatologists not aware of the clinical aspect of CBM or the most suitable mycological test.”
Reviewer 3 Report
Comments and Suggestions for Authors
This is a useful contribution at a time when there is an accepted need to increase information on the distribution of chromoblastomycosis
The distribution in the country is interesting, Is there anything specific or unusual about the geographic or environmental features of the endemic zone ?
Fumigoid cells. It would be helpful to provide the other more commonly used synonyms as well eg muriform cells, sclerotic bodies, medlar bodies etc
Strictly speaking these are not yeast cells. They may appear yeast like but it is better to describe them as parasitic phase cells. Dong et al Am J Trop med Hyg 2020 103 704-12
Surgical excision . How often should this be combined with an antifungal drug
Antifungals. Can you give some details of the doses used ?
Comments on the Quality of English LanguageFine.
Author Response
Reviewer 3
This is a useful contribution at a time when there is an accepted need to increase information on the distribution of chromoblastomycosis
The distribution in the country is interesting, Is there anything specific or unusual about the geographic or environmental features of the endemic zone ?
- This is a very interesting question. In our opinion, it probably reflects the importance of agriculture in the coastal region, though we cannot explain why traditional agriculture in Amerindians and Maroons in the hinterland does not lead to CBM cases. The deep Amazon rainforest in the French Guiana hinterland might be less suited for CNM, though Lobomycosis and Paracoccidiodiomycosis are known to thrive in this environment.
- See Discussion, page 18, lines 278-283
Fumigoid cells. It would be helpful to provide the other more commonly used synonyms as well eg muriform cells, sclerotic bodies, medlar bodies etc
- We agree with the reviewer. The synonyms have been added, please see Introduction, page 3, lines 68-69. “also called muriform cells, sclerotic bodies or medlar bodies.”
Strictly speaking these are not yeast cells. They may appear yeast like but it is better to describe them as parasitic phase cells. Dong et al Am J Trop med Hyg 2020 103 704-12
- We concur. The term “yeast” has been replaced with “parasitic phase cells” throughout the document. This reference has been added in the Introduction.
Surgical excision . How often should this be combined with an antifungal drug
- In our opinion, surgical excision should be performed when the lesions are superficial and limited enough not to expose bone or tendon structures, this is now explained in the Discussion.
Antifungals. Can you give some details of the doses used ?
- The details have been added to the Results, page 15, lines 227-228:” Itraconazole (200 mg to 400mg daily), 5-FC (12 to 24 tablets daily) and terbinafine (1g daily) were the most frequent antifungals.”